# A Qualitative Study of the Feasibility and Acceptability of Implementing ‘Sit-To-Stand’ Desks in Vocational Education and Training

**DOI:** 10.3390/ijerph18030849

**Published:** 2021-01-20

**Authors:** Mara Kirschner, Rianne H.J. Golsteijn, Sanne M. Sijben, Amika S. Singh, Hans H.C.M. Savelberg, Renate H.M. de Groot

**Affiliations:** 1Faculty of Educational Sciences, Open University of the Netherlands, 6419 AT Heerlen, The Netherlands; rianne.golsteijn@ou.nl (R.H.J.G.); sannesijben@hotmail.com (S.M.S.); renate.degroot@ou.nl (R.H.M.d.G.); 2Mulier Institute, 3584 AA Utrecht, The Netherlands; a.singh@mulierinstituut.nl; 3Department of Nutrition and Movement Sciences, School of Health Professions Education (SHE) and School of Nutrition and Translational Research in Metabolism (NUTRIM), FHML, Maastricht University, 6229 GT Maastricht, The Netherlands; hans.savelberg@maastrichtuniversity.nl

**Keywords:** qualitative research, sit-to-stand desks, vocational education and training, sedentary behaviour, focus group interviews

## Abstract

While it has been shown that interrupting a person’s sedentary behaviour has the potential to improve cognitive, physical and mental health, a large part of time that students spend in school is sedentary. As research has shown that approximately 80% of vocational education and training (VET) students have an unhealthy sedentary lifestyle, implementing “sit-to-stand” (StS) desks could interrupt sedentary behaviour and promote healthier behaviour. Therefore, the acceptability and feasibility of using such desks in the VET setting should be investigated. Using semi-structured focus group interviews analysed via deductive content analysis, the opinions of 33 students for the following topics were assessed: (1) usage of the standing option of the desks (2) reasons for standing in class (3) experienced effect of standing behind the desk, and (4) fostering future StS desks usage. Although VET students are aware of the potential benefits of using StS desks, they need to be actively stimulated and motivated by teachers to use them. In addition, time is needed to get into the habit of standing. Thus, for successful implementation of StS desks in the VET setting, all stakeholders (i.e., students, teachers, schoolboards) should be actively involved in stimulating the healthy behaviour of VET students.

## 1. Introduction

The large amount of sedentary time spent by adolescents is a worldwide problem causing severe health risks. According to the World Health Organisation, sedentary behaviour is a leading risk factor for non-communicable diseases and death worldwide [1]. This seems to be independent of a person’s physical activity level or age [2]. Sedentary behaviour is the time that is spent sitting, lying or reclining, with low energy expenditure (≤1.5 MET) [3]. In their systematic review, Tremblay et al. [4] showed that high amounts of sedentary behaviour are associated with metabolic syndrome, cardiovascular disease, decrease in physical fitness, increased depressive symptoms, lower self-esteem, decreased perception of self-worth and behavioural problems in children and youth. There is also growing evidence that a high amount of sedentary behaviour has a negative effect on cognitive and school performance [5,6,7]. 

Sedentary behaviour significantly increases from childhood into adolescence [8]. In the Netherlands, the self-reported amount of time spent sedentary (i.e., seated) among teens (12–18 years old) is 9.6 h a day [9]. Of these 9.6 h, on average, 3.3 h are spent seated in school, as teens primarily follow lessons where sitting is the standard. This is an important contribution to the total daily sedentary time [10]. A way to decrease this classroom-related sedentary school behaviour is to replace traditional desks with ‘sit-to-stand’ (StS) desks. These are desks that can be altered from traditional desks to standing desks using just a lever. However, before such desks are implemented, the question arises as to whether replacing traditional desks in the classroom with StS desks is feasible for students. Therefore, it is necessary to gain insight into students’ perception and the user experience of such desks. 

Interrupting sedentary time has more than just health effects [4]. An increasing amount of research shows that interrupting sedentary behaviour by using standing desks could have beneficial cognitive, physical and psychological effects for students. With regard to cognitive effects, Rosenbaum, Mama, and Algom [11] found a significant increase in selective attention in the standing, as opposed to the seated condition in two studies (*n* = 7 and *n* = 50). Another pilot study with high school students (*n* = 41) found that the use of standing desks was associated with a 7% to 14% improvement in cognitive functioning across several executive functioning and working memory tasks [12]. A reason for this could be that while standing, there is increased arousal in brain regions that are important for learning, compared to being seated, but there is no clear evidence on this yet [13]. In addition, Mazzoli et al. [7] found that a higher amount of sedentary behaviour resulted in more lapses in attention (*n* = 149). Perceived positive effects on attention and focus while using standing desks compared to traditional desks were reported in several focus group interviews among primary and secondary school students, as well as teachers [14,15,16].

Research also shows that interrupting sedentary behaviour in educational settings by using standing desks can have several physical benefits. Too much sedentary behaviour decreases energy expenditure and can potentially lead to weight gain and obesity over time [17]. A pilot study (*n* = 58) showed that energy expenditure increased by 17% in a classroom with StS workstations compared to a traditional classroom setting [18]. Moreover, a significant increase in energy expenditure for students in the intervention group was found by Reiff et al. [19]. Even though these findings are very promising, it should be taken into account that in both of these studies, the sample size was small. Furthermore, sitting for long periods at a time can cause musculoskeletal discomfort such as back and neck pain [20,21]. Research by Ee et al. [22] found a significant reduction in neck discomfort in the standing condition, and using the standing desks reduced the likelihood of reporting musculoskeletal discomfort in the neck, shoulders, elbows and lower back. In a pilot study (*n* = 30) by Hinckson et al. [23], primary school students reported no to little musculoskeletal pain or fatigue when using standing desks.

Too much sedentary behaviour is also associated with negative psychological effects. A review on sedentary behaviour and mental health in youth (10–19 years old) reports that high amounts of sedentary behaviour are associated with more depressive symptoms and psychological distress [24]. Although no research has been done yet on what interrupting sedentary time in the educational setting does for mental wellbeing, research done by Ellingson et al. [25] showed that a reduction of 60 min in screen time, which is often used as a proxy for sedentary time, significantly improved mental wellbeing. As sedentary time in school is not limited to screen time, it is uncertain if interrupting sedentary time increases mental wellbeing in the school setting.

Although interrupting sedentary behaviour has several benefits, and StS desks could be a tool to achieve this, benefits are only experienced if students actually use them in a standing position. Therefore, it is important to know what students’ attitudes are towards standing at StS desks. In previous studies, using focus group interviews, primary and secondary school students reported that they enjoyed using standing desks [16,23]. Benzo et al. [26] showed that more than three quarters (77% out of 993 students) were in favour of introducing standing desks (i.e., thus not StS desks) and indicated that they would stand 25–50% of the class time. University students reported that using standing desks would improve their physical activity and in-class attention, and decrease restlessness. This suggests that students probably would appreciate the opportunity to stand for at least a portion of their class time. However, interrupting sedentary behaviour is also perceived as challenging, due to peer pressure (i.e., imposing a certain type of behaviour on a peer), group norms (i.e., underlying attitude shared by peers), behavioural modelling (i.e., seeing a peer perform a certain behaviour can motivate others to do so as well) and co-participation (i.e., participating in behaviour with peers and contributing to behavioural reinforcement this way) [27]. Peer influence is mentioned in a study by Sherry [28] as a reason not to stand. In this study among primary school children, a rotating desk system was used, and five students mentioned being reluctant to stand because the rest of their peers were seated. In a study among primary and secondary school students by Verloigne et al. [15], its focus group interviews showed that more support from their teacher would probably have an effect on how much they would use the standing desks. Teachers also suggested that implementing enough desks for all students is important to ensure exposure to the desks and increase possibility to use the desks, since several previous studies replaced just a few desks and used a rotation system [15,16,23,26].

Although research has been done on the effects and applicability of standing desks in primary school [15,29,30], secondary school [15,16,31], and academic tertiary education settings [19,32,33], insight into their use in the vocational education and training (VET) setting is lacking. According to Rijpstra et al. [34], 80% of VET students have an unhealthy sedentary lifestyle and, thus, interrupting this sedentary lifestyle could also have significant beneficial effects for them. In addition, VET students are at a different life stage compared to primary and secondary students. They are educated for a broad diversity of specific vocational jobs and introduction into the labour market. Therefore, they work for a great part of their study. This makes that students at VET schools differ from students in other school settings, such as secondary schools and universities. Thus, generalising the results of earlier mentioned research is not possible. Furthermore, in the majority of the previous studies [15,16,23,26,29,30], only a few standing desks were implemented in a classroom, thus disabling getting a good view on what happens when you facilitate a whole classroom with StS desks. This is why, in the current study, we aimed to explore the feasibility and acceptability of implementing StS desks in VET classrooms.

## 2. Materials and Methods

### 2.1. Study Design

The current study had a qualitative research design. Focus groups with a semi-structured interview method were conducted to obtain in-depth information from students on the use of StS desks. A deductive approach was used to analyse the data [35]. The main themes used were themes created by Verloigne et al. [15]. While analysing the qualitative data, additional themes were created when data did not fit in the already existing themes.

### 2.2. Sample

The recruitment of participants was done using a purposive sampling method [36]. A VET school in the central part of the Netherlands was approached and asked to participate in the current research. In deliberation with the school management, 48 students, divided over two classes, were approached. Inclusion criteria for the students were: first year VET students, participating in a course ‘all round beauty specialist’ who never used StS desks in school before. The two specific classes were chosen by the school management, based upon the fact that for the duration of the study, they would be at school taking regular classroom classes. 

### 2.3. Procedure

The school management provided a list of teachers who taught the two participating classes. These teachers were invited to an information session to inform them about the content of the project, the goal of our research, the role they had within the research, and the advantages of standing versus sitting. After the presentation, we provided teachers with written information about the research.

Students were provided with information on the project in the form of a presentation given by the researchers. The presentation contained information on the practical use of the desks and the procedure of the study. It was emphasised that participation was voluntary, all data would be anonymised and stored securely, and that students could opt out at any time. No information was given on the presumed advantages or disadvantages of StS desks. At the end of the presentation, all potential participants received written information and were provided with informed consent forms, which could be handed in at the latest before the start of the focus group interviews (i.e., 3 weeks later). In accordance with Dutch ethical regulations [37], participants who were 16 or 17 years old had to inform their parents, by giving them the information letter, but could decide themselves if they wanted to participate. Participants 18 years or older decided themselves on study participation. The study was approved by the Ethical Research Board of the Open University in the Netherlands as an addendum on PHIT2LEARN, cETO ref: U2018/09408/HVM.

Before the start of this study, all traditional desks were removed from the two classrooms and replaced with StS, which were only used for the duration of this intervention. All lessons took place in these two classrooms for the duration of the study (3 weeks; September 2019–October 2019). During the first week, teachers did not advocate using the desks in a standing position, to see if solely having the StS desks at the students’ disposal would result in using the desks. This resulted in the desks barely being used for standing. Thus, in the second and third week, teachers were asked to stimulate students to use the desks in a standing position (i.e., by asking students to stand up in class). 

After three weeks of StS desk use, students were invited to attend semi-structured, in-depth focus group interviews. To ensure privacy and facilitate that students could talk freely, focus groups were conducted in a separate classroom. All focus groups were audio-recorded with two voice recorders (TASCAM field recorder DR 40X), to make sure that all participants were recorded clearly and that no data was lost for transcription afterwards. During the interview open, in-depth questions were asked, to obtain as much information as possible. In addition, students were asked to elaborate on the answers given. When the interviewer noticed that a question did not initiate a discussion, the interviewer addressed each individual participant for their view on the subject. This way, all participants were stimulated to provide information.

### 2.4. Focus Groups

All students who signed the informed consent were randomly assigned to focus groups. Those who did not sign the informed consent could use the StS desks during the three weeks, but were not interviewed. 

According to Masadeh [38], a focus group of four to six participants is optimal to get as much information as possible. We estimated that, for this number of participants per group, we would need roughly 60 min per focus group. 

A semi-structured interview guide was developed based on the interview protocol of Verloigne et al. [15]. This protocol was used as an example, since they conducted a similar study in primary and secondary schools in Belgium. The focus group started with an explanation of how the interview would be conducted, how the obtained information would be used, and by emphasising that the participants would be anonymised in the transcripts. After this, an introduction round was used as an icebreaker. Then, the main questions for each category were asked: (1) usage of the standing option of the StS desks; do students use the standing option, and what triggered them to use or not use it, (2) reasons for using the StS desks independent of actually using it; can students think of reasons why they would or would not use the desks in a standing position, (3) experienced effect; what do students notice when using the standing option of the StS desks, (4) fostering future StS desks standing; what do students need to use the standing option of the desks in the future. Follow-up questions were asked to elaborate on the answers given by the students. 

### 2.5. Data Collection

During the focus groups, two researchers were present, one acting as an interviewer and the other acting as an observer. The interviewer conducted the interview, while the observer made sure all questions were adequately addressed (i.e., by checking the questions in the used protocol, reminding the interviewer in case of missed questions) and making notes of disruptions during the interview protocol.

The interviewer for all focus groups was the first author (M.K., female researcher). Students had seen the interviewer once before, during the introductory presentation. Before conducting the focus groups, M.K. pilot tested the interview protocol with colleagues, to get acquainted with the questions that needed to be asked. During the first three focus groups, the observant was another female researcher, who also met the students once prior to the focus groups during the introductory presentation. During the last three focus groups, a female master student who was a former teacher at the school was the observer. 

### 2.6. Data Analysis

To analyse the data, deductive qualitative content analysis was used. The data were analysed according to Cho et al. [39]. All audio-recordings were transcribed and afterwards scored in EXCEL files. Each participant in a focus group was given a number. In cases where it was unclear from the recordings who was speaking, the speaker was coded as unknown (UK). For scoring, a coding scheme was created (see Figure 1). The main themes in this coding scheme were derived from research done by Verloigne et al. [15], as explained in the focus group paragraph. Afterwards, subthemes were created from the used interview protocol, and a preliminary code scheme was constructed. Thereafter, subthemes were adjusted and added to the coding scheme, based on what was said during the focus groups, which resulted in the final coding scheme. Lastly, the content of the focus group transcripts was assigned to the different themes and subthemes. 

Three focus groups were independently coded in EXCEL by two team members using the final coding scheme. To assess the interrater reliability (IRR), a Cohen’s Kappa was calculated. A Cohen’s Kappa score of 0.60–0.79 is seen as moderately reliable (35–63% agreement), a score of 0.80–0.90 is seen as strong (64–81% agreement), a score of above 0.90 is seen as almost perfect (82–100% agreement) [40].

### 2.7. Coding Scheme

During the interview, the four main themes gave structure to the interview. When creating the coding scheme (see Figure 1), the interview protocol was used to create subthemes per theme, and the transcripts were used to finalise the subthemes as described in the methods.

Theme 1 (i.e., usage of StS desks), was centred around the answers on the question whether students used the standing option of the StS desk and why. Generally, students did not elaborate on their answers when asked. Theme 1 was divided into negative/positive responses. Which could either be instruction form dependent, lesson dependent, or usage because of the part of the day. A response could also be coded as neutral when students only told the interviewer whether they tested the desk or did not use the option to stand of the desk at all, without any other explanation.

Theme 2 (i.e., reasons for using the StS desks) was based on the question of whether the students could think of reasons to use or not use the StS desk in a standing position. Answers could be divided in either stimulating or obstructive reasons to stand. The subthemes could either be cognitive (e.g., concentration), physical (e.g., muscle/joint), psychological (e.g., shame), practical (e.g., giving presentations).

Theme 3 (i.e., experienced effects of the StS desks) was based on the question what the students experienced while standing. Answers could be divided into negative/positive responses and subthemes could either be cognitive (e.g., concentration), physical (e.g., muscle/joint), psychological (e.g., shame), practical (e.g., desk stability), or a neutral response when no effects were experienced.

Theme 4 (i.e., fostering future StS desks usage) was based on the question of what students needed to start or keep using the stand option of the desks. Answers given by the students were based on in-class logistics, either being instruction form-dependent (e.g., rules), class setup (e.g., standing people in the back), practical (e.g., desk stability) or reinforcement (e.g., reward for usage).

## 3. Results

### 3.1. Participants

A total of 48 students were approached to participate and 43 agreed as indicated by their (and if applicable, their parents’) signed informed consents. These students were divided into six focus groups. Because of a technical error with the voice recorders, the data of one focus group consisting of 6 participants was lost, 4 participants who signed an informed consent were absent on the day of data collection. This resulted in data for 33 participants in the data analysis. All non-responders were female. 

These 33 participants, 32 females and 1 male (this distribution is representative for this study track) with a mean age of 17.3 years old (SD = 2.2), ranging from 16 to 28 years old, were divided over the six focus groups. Due to absent and/or late students, focus groups needed to be rearranged on-the-fly. This resulted in group sizes of three participants (1 focus group), five participants (1 focus group), six participants (3 focus groups), and seven participants (1 focus group). Although one group consisted of only three participants due to reasons mentioned above, the data were assumed to be valuable and were thus transcribed and coded.

### 3.2. Interrater Reliability

The coding of the first three transcripts by two coders resulted in strong IRRs of respectively 0.85, 0.78 and 0.82. As all three transcripts had a strong IRR, the remaining three transcripts were only coded by the main researcher. For the consistency of the data, the main researcher’s first three coded transcripts were used.

### 3.3. Theme 1: Usage of StS Desks 

Students often mentioned not standing at the desk at all, and only a few students tested the stand option of the StS in the first week when they did not receive any instructions to use the desks (see Table 1). During the second week when teachers started to motivate and instruct students to do so, more students tested the desks. When students talked about just testing the desks, students said things like: “We tested it once with the whole class, nobody liked it” (Student (ST) #4, Focus Group (FG) 6). After the second week, students mentioned using the desks more often in a standing position because teachers stimulated them to do so. “Not the first week, but the second week” (ST #UK, FG 4) and “We didn’t stand a lot, only every now and then” (ST #UK, FG 5). Some students mentioned not standing behind the desks at all, because they were used to sitting in class. “Sitting is normal and because we were in the same classroom the whole time, you just sit down. You don’t really remember to stand.” (ST #UK, FG 5).

When students did test and/or use the StS desks in the standing position, they often mentioned that it was because the teacher instructed or motivated them (instruction form-dependent). For instance, teachers instructed the students to use the desks in the standing position during Kahoot^®®^ (a multiple-choice quiz done in a group): “During Kahoot I liked standing, but you also notice that an active stance is suited for playing Kahoot and you’re doing it with the whole class, which makes it suitable for a group activity” (ST #UK, FG 3). Teachers also just told them to use the desks in the standing position “Because it was mandatory” (ST #UK, FG 4) or teachers motivated the students to use the desk: ”During Dutch class, when the teacher asked us to stand” (ST #UK, FG 2). During some specific classes (i.e., citizenship class), students mentioned trying the StS desks; it seems that during these classes, they often have to give presentations, so having a desk that can be put in a standing position is beneficial, “Yes during citizenship class.” (ST #UK, FG 5), “Only during my Dutch class presentation” (ST #4, FG 2). The subtheme ‘part of the day’ (i.e., morning, afternoon) was derived from the protocol; students did not mention this as a reason to use the StS desks.

Although students did not use the desks in a standing position that much, they could not explain why they did not. In some cases, the instructions were counterproductive (instruction dependent), and students did not want to stand because they were told to do so: “Everybody had to stand according to the teacher, but some just really didn’t want to” (ST #UK, FG 5). One student mentioned that it was not possible because of a lesson (lesson-dependent) “With algebra it wouldn’t be possible, I think” (ST #UK, FG 3). Students mentioned that they did not use the desks because they were too tired to stand in the morning (part of the day). They mentioned 8:30AM as being too early to already stand behind their desks “But at half pas eight I really don’t want to stand” (ST#7, FG1).

### 3.4. Theme 2: Reasons for Using the StS Desks

To the question “What could be reasons to use or not to use StS desks in a standing position?”, students reported stimulating reasons, as well as obstructive reasons that they could think of, but were not necessarily applicable to themselves or that they had experienced (see Table 2). Students mentioned some stimulating reasons that could be relevant to start standing behind a desk. These reasons were either cognitive (e.g., regaining attention), physical (e.g., better for posture), psychological (e.g., a feeling of autonomy) or practical (e.g., easier to talk to each other). Being more concentrated and alert while standing, was a cognitive stimulating reason mentioned by the students: “Well, you’re more awake when you stand, I think” and (ST #UK, FG 2) “When you stand for ten minutes, your concentration is better” (ST #UK, FG 5) and “Because you show more on-task behaviour” (ST #UK, FG 4).

Almost all students mentioned physical stimulating reasons to stand up in class. For instance it enhanced an active posture: “You’re not able to sit slumped in your chair, when you stand, that’s chill” (ST #UK, FG 3), and that it was better for their back “Maybe it’s better for our posture and our back. Because while sitting we’re often bent forwards. Now we can also stand up straight.” (ST #8, FG 1).

Psychological stimulating aspects that were mentioned are, first, that students liked the feeling of being able to decide themselves what to do; it gave them a feeling of autonomy: “On the other side it’s nice that you have some kind of freedom, you can do what you want” (ST #5, FG 3). In addition, students also mentioned that standing behind a desk could give a feeling of secureness “Yes, I think when giving a presentation, it feels safer. You can have something around you, so to say” (ST #7, FG 2).

Students also talked about practical reasons to use the StS desks in standing position. They, for instance, said that the desks could be used for giving presentations: “For presenting its nicer, I think” (ST #UK, FG 1). In addition, they mentioned, as a practical reason, that it was easier to see each other and interact with each other. “And because you have a better view, it’s easier to look at people” (ST #UK, FG 2).

The obstructive reasons (e.g., reasons not to use the StS desk in a standing position) were categorised in the same four subthemes as the stimulating reasons (cognitive, physical, psychological and practical). The only cognitive obstructive reason mentioned was that standing up could distract other students: “Because I don’t want to distract others” (ST #UK, FG 4).

Students came up with a number of physical barriers. For instance, using the StS desks in a standing position could hurt their muscles and joints “And when standing, then you’ll probably get a hollow or arching back, or you stand on one leg” (ST #UK, FG 4), “My feet hurt, some people get back pain, and some people become dizzy” (ST #UK, FG 6). But another physical barrier that was mentioned was that standing was tiresome: “Standing is pretty tiresome” (ST #UK, FG 5), “We’re lazy” (ST #UK, FG 1).

Students could also imagine some psychological barriers to not stand behind a StS desk. They do not want to be the centre of attention and having the group norm of sitting in class also makes breaking the habit of sitting in class hard. When they mentioned standing could be experienced as being the centre of attention, they said things like: “Yes, it looks as if you’re presenting, as if you stand in front of the class, and say look at me, I’m standing so I have something to say” (ST #UK, FG 4). Insecurity was also mentioned as a possible reason not to stand, “So, maybe we’re too insecure to stand” (ST #UK, FG 4) and “Being with 18 classmates in a classroom, it’s a kind of awkward because everyone is looking at you” (ST #7, FG 2). 

Students also mentioned being concerned about not being able to see the front of the class, students that are standing in class could block the view of the student that is seated behind them (practical), “You could stand in somebody’s sight” (ST #UK, FG 3) and “But when it happens in front of your nose, you know what I mean? (ST #UK, FG 5). 

### 3.5. Theme 3: Experienced Effects of Using the StS Desks

The experienced effects could either be positive, neutral (e.g., no difference noticed between sitting or standing) or negative (see Table 3). Students’ experiences were further categorised into either cognitive, physical, psychological or practical subthemes. Students mentioned relatively few positive cognitive effects. One student mentioned that it regained her concentration: “You immediately concentrate again” (ST #UK, FG 5) and another girl mentioned that standing did not distract her, “I didn’t find it as distracting as X” (ST #UK, FG 4). 

Students agreed on experiencing positive physical effects while using the desks, adjusting the desks to the perfect sitting height felt good for their posture, so it should be noted that these students did not use the StS desks in a standing position: “You could put it a bit higher, causing you to sit up straight, I liked that. I had less back pain” (ST #7, FG 3), “I’m a little bit smaller, so the possibility to put it a little bit lower, made that I could sit comfortably” (ST #8, FG 2). 

Regarding positive psychological effects, some students mentioned that it felt nice to stand every once in a while; it was categorised a positive feeling due to standing and thus categorised as psychological, “It’s just nice to stand every once in a while” (ST #7, FG 1), “Standing is also nice sometimes” (ST #U,2FG 5). 

The positive practical effects that students talked about were more or less the same as the physical positive effects that they mentioned. The capability of the table to being adjustable in height “I liked that you can adjust it in height” (ST #UK, FG 1) and being able to read notes while presenting “Also that when you have to read something for the class, and that you so to say have it on eye height, instead of having to look down, because that makes it harder to also look at the class” (ST #8, FG 2).

When mentioning negative effects, students discussed some cognitive factors such as feeling distracted when standing “I did try it, when we had to, but for me I prefer learning while seated, I’m able to concentrate better and keep my focus” (ST #5, FG 3), and “It’s distracting” (ST #UK, FG 5) were mentioned by the students. 

When talking about negative physical effects, students mainly mentioned that standing caused muscle and joint pain: “Because my legs hurt, and my feet hurt” (SP #UK, FG 4), “Yes and my lower back hurts very fast ” (ST UK, FG 4), and it was getting tiresome after a while “Yes, well I just don’t like standing to be honest, not chill at all, I also get tired legs, so yes, after two minutes I’ll sit down” (ST #6, FG2). 

When discussing negative psychological effects, the main effect mentioned was that it just did not feel nice “But because everyone around me was sitting, it didn’t feel nice to be standing” (ST #2, FG 3), “it just does not feel nice, to be standing the whole time” (ST #5, FG5). In addition, the students reported that they had the feeling people were looking at them while standing, “Sometimes people did stand, but then the whole class started looking like, oh the table is going up” (ST #UK, FG 4), “It’s different when everybody stands than when you are the only one” (ST #6, FG 4) “It was funny, but eventually you are like, well, my friends are sitting too, and why am I standing?” (ST #UK FG2). 

The negative practical reasons mentioned had to do with the classroom being messy, “I thought it was messy, the chairs are in the way, behind you so…” (ST #4, FG 4), “And bags” (ST #UK, FG 4), “Yes, it’s also a bit unorganised, it looks very messy” (ST #3, FG 6). One student also mentioned that standing provoked walking: “Yes, you also tend to walk around the classroom. When I sit, I sit, but when I stand you just go and walk easier” (ST #UK FG 3). Out of all mentioned practical negative effects, the aspect that was mentioned most frequently was that the tables were far from ideal for these students to use: “When you want to put the desk back down, you really have to hang over the table, at least when you do not have that much power” (ST #UK FG 5). Students also mentioned that the tables were unstable, “Just the wobbling and stuff” (ST #UK FG 2), “And then I stood and the table was very much wobbling back and forth the whole time while typing, so I quickly sat down again” (ST #6 FG 1), “The foot in the middle is also annoying” (ST #UK FG 4).

### 3.6. Theme 4: Fostering Future Implementation of StS Desks

On the question what students needed to start using the desks, they mentioned that they needed teachers to give them instruction on when to stand (see Table 4). Teachers should keep the class setup in mind, and it is important that desks are stable, sturdy and easy to use. When students talked about their need for specific instructions, they said, for instance: “Well I think, when presenting, it’s nice to stand. So, when the teacher says, for instance: X you have to present, then you stand up” (ST #UK, FG1). Students also mentioned that standing could be actively integrated in the lessons: “Maybe just as an exercise, everybody stands up and after a while you can sit down again” (ST #UK, FG2), “Or during instructions” (ST #UK, FG5). It also seems important to students that teachers take an active approach and instruct and motivate students to use the desks “I think teachers should motivate us more to use the desks” (ST #UK, FG2), “maybe building it up, starting every lesson for instance with 10 min of standing and then sitting again, so people get used to it” (ST #6, FG3), “just oblige standing” (ST #UK, FG2).

When students mentioned the class setup as being an important aspect for implementation, they mentioned that desks should not be put in a U-formation. Instead, they preferred that the desks were put into rows, as well as putting the StS desk in the back. In this way, all students are still able to see the front of the class if they decide not to stand, “I think placement of the desks plays a big role in this, when being in a U-formation everyone can see you. When you put the standing desks in the back, and put them in regular classroom formation not everyone is able to see you, so you don’t draw that much attention when you want to stand up. So, I’d prefer that to using a U-formation.” (ST#UK, FG4) and “Maybe another setup, as X said, tables in the back for students who want to stand, and the students who want to sit in the front” (ST#UK, FG4). Moreover, the students mentioned that it could be motivating if the desks were already in standing position, when they entered the classroom: “I think if we enter the room and the tables were already in a standing position, I would stand instead of putting the desk down” (ST #UK, FG5).

Practicality was mentioned when the students talked about the stability and ease of use of the desks. “Yes, they have to become more robust” (ST #2, FG1). “Well I think they need to become more robust. They are not really stable, sometimes when you lean on it, you feel it moves. So, in my opinion they weren’t very robust” (ST #UK, FG2).

Two students mentioned that they should be compensated for using the desks. “They should shorten the lessons” (ST #UK, FG6), “Get a price for our tiredness” (ST 5, FG1).

## 4. Discussion

In this study, we aimed to gain insight into the perception of VET students on the use of StS desks in the classroom and what these students need to foster the use of these desks. Insight into four main themes (i.e., (1) usage of the standing option of the desks (2) reasons for standing in class (3) experienced effect of standing behind the desk, (4) fostering future StS desks usage) was gained using focus group interviews. Interestingly, students reported liking the feeling of being able to decide themselves when to stand behind the StS desks, but to actually use them in the standing position, the role of the teachers seemed to be most important. 

When VET students tried or used the StS stand desks, in our sample, they mostly stood behind their desks because teachers stimulated and motivated them. This is in line with research by Verloigne et al. [15] where students mentioned that they would stand behind the desks more if teachers motivated them to do so. A reason for this being reported by the VET students is that they are used to sitting in class, and thus forgetting that there is the possibility to stand. In this study, VET students also mentioned that when schools want to implement StS desks, teachers should play a leading role. 

Students mentioned potential benefits of using a StS desks, such as increased alertness, improvement of body posture, increased feeling of autonomy and practical benefits, for instance the desks being a good tool for giving presentations. This positive attitude towards using StS desks is in line with research done by Benzo et al. [26] in which students mentioned being willing to start using such desk if they were provided with one. Yet, despite students having a positive attitude, our findings also showed that they hardly stand behind the desks without stimulation from teachers. Thus, there seems to be a gap between having a positive attitude and using the desks, the so-called intention–behaviour gap [41]. A possible reason for this gap could be that students have the habit of being seated in class, and need more time and stimulation to break this habit and to get used to a new way of following lessons (standing instead of seated). 

VET students also mentioned that they experienced muscle and joint pain when standing during class. This differs from earlier research done by Ee et al. [22] and Hincson et al. [23], in which students reported less musculoskeletal discomfort. A reason for this difference could be that in these studies desks were replaced with stand biased desks, so students stood during all classes for a minimum period of 21 days. In the current study desks were not used standing very often. Another result of this study is that VET students mentioned that they got distracted by other students using the desks, which was also reported by Erwin et al. [16] and Verloigne et al. [15]. A reason for this could be that because the desks were not used very often, when they did put the desks in a standing position, it was distracting. 

In this study, when VET students discussed reasons not to stand behind their StS desks, they mention that they would be scared of being the centre of attention and not adhering to group norms. When discussing experiences while standing, they indeed confirmed the feeling of being the centre of attention. Not wanting to be the centre of attention and not complying to group norms was also mentioned by Sawka et al. [27] and Sherry [28] as a reason for students not to use standing desks. 

Since Hoare et al. [24] and Ellingson et al. [25] found an association between sedentary behaviour and mental wellbeing, it was expected that students would report improvements in mental wellbeing. However, they did not mention an improvement in mental wellbeing, nor did they mention wellbeing as a possible benefit of standing. A reason for this could be that an improvement of mental health is only noticeable with substantial decrease in sedentary behaviour, as found by Ellingson et al. [25]. The actual use of the StS desks in our study was limited and confined to a maximum of three weeks. This could be a reason for not experiencing an improvement in mental wellbeing. Another reason could be that students were not aware that an improvement in mental wellbeing could be a benefit of standing. What consequently could have resulted in not consciously experiencing an improvement in mental wellbeing and thus not reporting it as an experienced benefit of standing. In addition, another reason that improved mental wellbeing was not reported, could be that talking about mental wellbeing is considered by students as a too sensitive subject to discuss in a focus group, and was thus avoided. 

Practically, students were worried about not being able to see the front of the classroom when students in front of the class are standing. They mentioned that rearranging the classroom could be a solution for this. Students also mentioned another practical disadvantage, the desks felt unstable and not user friendly, what prohibited them from using the desks more often in a standing position. Thus, in our study, easy usability of the desk seemed to be important, yet according to our knowledge, none of the previous studies reported anything regarding usability of the desk. 

Thus far, no research had been carried out on the acceptability and feasibility of StS desks in the VET setting. As indicated in the introduction, VET students differ from previously studied student populations in several ways. However, our results showed that overall VET students reported benefits and nuisances similar to other populations. Yet, in contrast to primary and secondary school student [22,23], VET students seem to experience muscle and joint pain from using the desks and not relief. It is not clear whether this difference is related to the study population or to differences in the intervention itself, as discussed previously. In addition, VET students seem to be unique in mentioning the importance of the stability and usability of such desks. It is not clear if this, for instance, is because of the type of desk that was used or because as students grow older the desks are used more extensively (e.g., to put laptops, study books and notebooks on) and thus the usability of desks becomes more important, although that would also be the case for tertiary higher education students, and therefore not only relevant for VET students. Thus, despite the specific characteristics of VET students, these did not result in substantially different outcomes regarding acceptability and feasibility.

### 4.1. Suggestions and Recommendations

As our study revealed, students experience difficulties in using StS desks. To stimulate the successful use and application of StS desks in VET classrooms, some suggestions and recommendations for this can be made. We recommend, first, that teachers provide structure to students on when to use and when not to use the StS desks. This can be achieved, for example, by having the desks in a standing position before students enter the classroom and telling students to stand during a specific part of the lesson (e.g., when giving instructions or for group work). Something else that could stimulate use, is implementing StS desks from primary school onwards, as using such desks and thus standing during class would become the norm for students. Because this suggestion will only establish results on the long term, a more short-term solution could be to provide teachers with ‘tips and tricks’ on how to best use StS desks in their lessons. A solution to the feelings of group norms that inhibited some students using the StS desks might be to use the desks in groups. Teachers could, for example, ask all students to stand, while instructing about a certain task or while students are carrying out group exercises. With respect to StS desks being perceived as a distraction (i.e., when changing the height of the desks), a solution could be to use StS desks for an extended period of time. This could lead to desensitisation for distractions of the desks being used in either the sit or stand position. Another possibility could be using a different type of desk (e.g., stand-biased desks, which are always in a standing position and can be used with a barstool). In order for students to overcome their experienced discomfort in muscles and joints, standing time should be built up gradually, progressively increasing the amount of time standing and decreasing the amount of time sitting. 

### 4.2. Strengths and Limitations

In the current study, the classrooms were equipped with only StS desks, so all students could use these desks whenever they wanted without having to wait until desks were available. This way, all students had the opportunity to try and use the StS desks whenever they wanted, and thus experience all potential benefits and drawbacks of such desks, allowing them to provide valuable input regarding the use of these desks in the focus group interviews. This differs from previous studies in which classrooms were only equipped with a few StS desks [15,16,23,26,29,30]. Furthermore, the focus group protocol was pre-tested to ensure all topics were sufficiently covered, this resulted in the interviewer being confident and capable during the focus group interviews. This enabled the interviewer to create an open atmosphere, in which students were able to talk freely about their thoughts and ideas concerning the use of the StS desks. Lastly, a coding scheme was created to make sure the data could be analysed consistently. The calculated Cohen Kappa for the coding of the first half of the focus groups was high, thus indicating that the data was reliably analysed using the coding scheme. 

Due to lack of intrinsic motivation, the StS desks were not used in a standing position the first week. Therefore, it is possible that there was not enough repetition and stimulation for standing to become a habit. For habits to become behavioural responses, they need to go through a history of systematic repetition and reinforcement [42,43]. Future research should give students structure, cues or other reinforcements on when to use the desks, and use the desks for a longer period of time. Additionally, in the current study, teachers were not provided with an intervention protocol, because it was assumed that students would use the desks intrinsically. Due to the lack of standing behind the desks by students, teachers were ad hoc provided with examples on how to stimulate students and incorporate the desks within their lessons. This probably would have been more effective if this was developed in collaboration with the teachers. Furthermore, we had no objective insights on how often students stood in class. Possibilities to objectively measure standing could be done using accelerometery, using video registration of the classes, or asking teachers to score how often students stood behind their desks. However, we decided not to use such measurements, since they are burdensome and do not fit within the scope of the current study. Lastly, generalising the results of the current study should be done with caution, since this research was conducted at only one school and one track within the VET setting. In addition, the current study reflects predominantly the opinion of females. Although this might be a limitation regarding the generalisability of the results, it should be noted that the researchers were females as well. As literature has shown that adolescent girls seem to feel more comfortable talking about their own experiences and feelings with same-gender interviewers [44], this combination of interviewers and interviewees may have been beneficial for our findings.

## 5. Conclusions

This study showed that teachers play a crucial role in successful implementation of StS desks. Overall, VET students seemed to have a positive attitude towards the StS desks comparable to other school settings. However, this study showed that just having a positive attitude is not enough to start using the desks. For VET students to experience the cognitive, physical and mental benefits of using the StS desks, teachers were essential. Teachers were needed to give structure and motivation to stand behind the desks. This means that teachers should know how big their role is in implementing new behaviour in the classroom, and be aware that, for students, there are benefits of using StS desks to interrupt sedentary behaviour. This could also help in overcoming the psychological hurdle of not wanting to draw any attention by being the only one in class who is standing and thus not complying to the norm of sitting. Another important finding is that acceptability and feasibility can be increased by using StS desks that are of good quality and are easy to use, increasing confident use. In summary, it is important to actively involve school boards and teachers in playing an active role in stimulating and motivating students to stand behind their StS desks and thereby promoting a healthier lifestyle.

## Figures and Tables

**Figure 1 ijerph-18-00849-f001:**
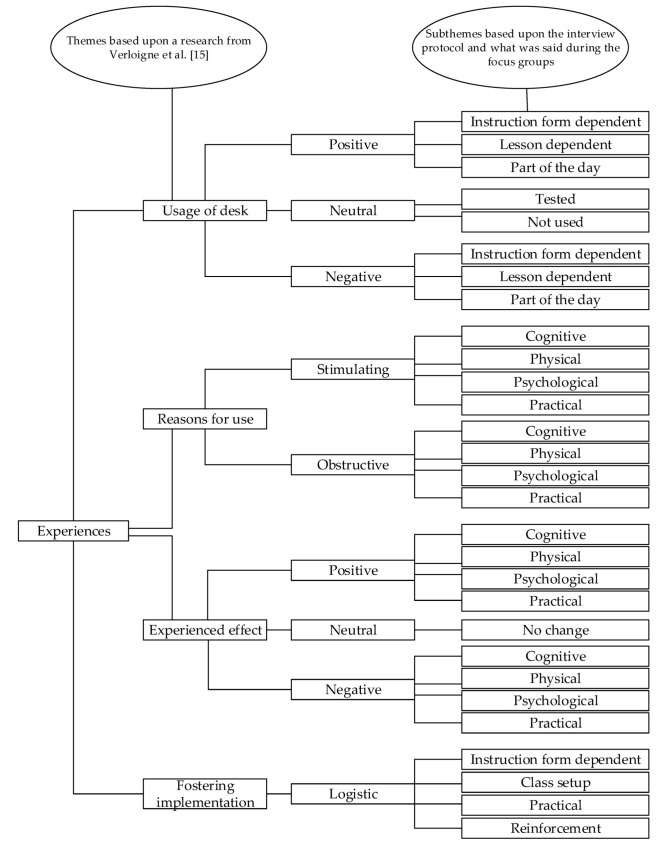
Coding scheme for evaluating the interview transcripts.

**Table 1 ijerph-18-00849-t001:** Summary of the results for theme 1: Did you or did you not stand behind your sit-to-stand (StS) desk and why?

Yes, with reason	Instruction form dependent	Teachers instructed or motivated students to stand
	Lesson dependent	For some specific lesson it was practical to stand behind the StS desks
	Part of the day	Not applicable
Yes/no, withoutreason	Tested	The standing position is tested
	Not used at all	Not used the standing position
No, with reason	Instruction form dependent	Instructions gave a counterproductive response, not standing
	Lesson dependent	For a specific lesson it was not practical to stand behind the StS desks
	Part of the day	The morning was too early to stand in class

**Table 2 ijerph-18-00849-t002:** Summary of the results for theme 2: What are reasons to use or not use the desk in a standing position?

Stimulating	Cognitive	Increase of alertness and on task behaviour
	Physical	Improvement of body position/posture
	Psychological	Increase of feelings of autonomy and safety
	Practically	Promoting communication and presentation
Obstructive	Cognitive	Distracting other students.
	Physical	Discomfort in muscles and joints, tiring
	Psychological	Scared of being the centre of attention, and not conforming to peer norms
	Practically	Blocking the view to the front of the class

**Table 3 ijerph-18-00849-t003:** Summary of the results for theme 3: What did you experience while standing behind the desk.

Positive	Cognitive	Not applicable
	Physical	Adjusting the height for a comfortable posture while seated
	Psychological	Nice feeling
	Practical	Being able to adjust the height of the desk
Neutral	No change	No difference noticed between a standing or seatedlesson
Negative	Cognitive	Loss of focus and concentration
	Physical	Discomforts like hurting feet and legs
	Psychological	Lack of conforming with the group norm and fear ofbeing the centre of attention
	Practical	Unstructured messy classrooms and unstable notwell-designed StS desks

**Table 4 ijerph-18-00849-t004:** Summary of the results for theme 4: What do you need to start using the sit to stand (StS) desk?

Instruction form-dependent	Support is needed from teachers in the form of instructions. This could be either with a specific task, during a specific part of a lesson (i.e., when course material was explained), or just for a few minutes at a specific point in time (i.e., beginning, middle or end of a lesson).
Class setup	Desk placement is an important factor to keep in mind when implementing the desks (i.e., u-formation vs traditional setup, putting standing desks in the back). All desks should be in standing position upon arrival.
Practical	Desks should be easy to use, feel stable and sturdy.
Reinforcement	Compensation for standing up in class is needed by having shorter lessons.

## Data Availability

The datasets analysed during the current study are available from the corresponding author upon reasonable request.

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
