# Peer review of "A Qualitative Study of the Feasibility and Acceptability of Implementing ‘Sit-To-Stand’ Desks in Vocational Education and Training"

_ijerph, 2021, doi:10.3390/ijerph18030849_

Round 1

Reviewer 1 Report

The study is introduced very well with several paragraphs which cited relevant studies on the topic (Lines 31 - 116). The reason for conducting the study is clearly given with a paragraph (Lines 117 - 131). The study design is clearly described and clearly fits a qualitative approach. How research ethics are respected is clearly described but a source should be cited for the Dutch ethical regulations regarding participants younger than 16 years (Lines 161 - 162). It will also be helpful to describe how the researchers interacted with parents of participants under 16 years to get consent (Lines 163 - 164).

It is not clear whether  the StS desks as mentioned in Line 169 were made available sorely for this study or whether they were already available but never used before. It appears the use of StS desks was under the control of teachers who would either give or not give instructions to students to use them (Lines 170 - 173; 287 - 288), as such 'optional StS desk use (Line 174) is confusing. It would be helpful to explain what is meant by 'teachers did not advocate .......' (Line 170 -171) and 'teachers were asked to stimulate students......' (Line 172 -173). Explain the importance of using two voice recorders (Line 174). Explain how participants were given turns in answering the questions (Lines 179 -180) and show how this would not intimidate some participants. Explain how the second researcher 'made sure all questions were adequately addressed' (Line 205) during data collection. Is it necessary to mention (Line 207) the gender of the interviewer and the observer (Line 211 and 213)? What is the role on an observer (Lines 211 and 213). Based on the optimal number of participants in a focus group (Lines 186 - 187), why were 3 participants (Line 270) referred to as a focus group? Explain the identification of  some participants as 'ST #Unknown(UK) while others are identified with a number such as 'ST #4'. Why would some participants be 'too tired to stand in the morning'?, I thought they should be fresh unlike in the afternoon.

The discussion is fairly presented. Explain your reasons for discussing something that 'students did not mention' (Lines 513 - 524). You only used literature for comparison; use literature also to explain the findings. Use previous studies to explain rather than to indicate that 'non of the previous studies reported.....' (Lines 532 533). You used speculation ('a possible reason...') and not literature. Your suggestions or solutions (Lines 489 - 492; 498 - 499; 503 - 505; etc.) are not coming from what participants said and are therefore your recommendations which should be moved to the conclusion and be added to what you recommended in Lines 583 - 586. 

Reviewer 2 Report

This was a qualitative study on the feasibility and acceptability of sit-to-stand desks in VET students. It is a very well written manuscript with sound background, rationale, methods and analyses.I've only minor spelling corrections/recommendations and one suggestion for adding to discussion (because it was claimed as one of the main reasons for conducting this research).

Affiliation: correct the affiliation of the author Renate H.M. de Groot (1 instead of 2)

line 73 – In this context, "intervention" might be a more appropriated term than "treatment".

line 79 – typo capital N (n=30)

line 97 – something is wrong in this part of the sentence "used a rotation system, this so exposure to and usage of the desks increases". Please, amend.

lines 98–99 – this sentence is very similar to the one in lines 91–92

line 170 – Although I’m not a native English speaker or writer I think that possession (‘s) is only used when a person possesses something not "things" possessing "things". I believe “study duration” or “duration of the study” are the correct forms.

DISCUSSION: I think that a paragraph contrasting your findings on VET students with the findings of other similar researches but conducted in students from other study cycles would be important because it was one of the main reasons for conducting this investigation (even if important differences in methods were noticeable, such as the number of desks, time of study, etc.).

CONCLUSION: I think the first sentence is unnecessary.

line 572 – Please consider placing VET before students: "Overall, VET students..."

Round 2

Reviewer 1 Report

You addressed comments adequately